# Usability of pulse oximeters used by community health and primary care workers as screening tools for severe illness in children under five in low resource settings: A cross-sectional study in Cambodia, Ethiopia, South Sudan, and Uganda

Theresa Pfurtscheller[1,2]*, Kevin Baker[1,2], Tedila Habte[3], Kévin Lasmi[1], Lena Matata[4,5], Akasiima Mucunguzi[6], Jill Nicholson[2], Anthony Nuwa[6], Max Petzold[7], Mónica Posada González[8], Anteneh Sebsibe[3], Tobias Alfvén[1,9], Karin Källander[1,10]

1 Department of Global Public Health, Karolinska Institutet, Solna, Sweden, 2 Malaria Consortium, London, United Kingdom, 3 Malaria Consortium Ethiopia, Addis Ababa, Ethiopia, 4 Swiss Tropical and Public Health Institute, Basel, Switzerland, 5 Malaria Consortium South Sudan, Aweil/Juba, South Sudan, 6 Malaria Consortium Uganda, Kampala, Uganda, 7 School of Public Health and Community Medicine, University of Gothenburg, Gothenburg, Sweden, 8 Malaria Consortium Cambodia, Phnom Penh, Cambodia, 9 Sach's children and youth hospital, Stockholm, Sweden, 10 UNICEF, New York, New York, United States of America

* resa.pfu@gmail.com

## Abstract

Timely recognition and referral of severely ill children is especially critical in low-resource health systems. Pulse-oximeters can improve health outcomes of children by detecting hypoxaemia, a severity indicator of the most common causes of death in children. Cost-effectiveness of pulse-oximeters has been proven in low-income settings. However, evidence on their usability in community health settings is scarce.This study explores the usability of pulse-oximeters for community health and primary care workers in Cambodia, Ethiopia, South Sudan, and Uganda. We collected observational data, through a nine-task checklist, and survey data, using a five-point Likert scale questionnaire, capturing three usability aspects (effectiveness, efficiency, and satisfaction) of single-probe fingertip and multi-probe handheld devices. Effectiveness was determined by checklist completion rates and task completion rates per checklist item. Efficiency was reported as proportion of successful assessments within three attempts. Standardized summated questionnaire scores (min = 0, max = 100) determined health worker's satisfaction. Influencing factors on effectiveness and satisfaction were explored through hypothesis tests between independent groups (device type, cadre of health worker, country). Checklist completion rate was 78.3% [CI 72.6–83.0]. Choosing probes according to child age showed the lowest task completion rate of 68.7% [CI 60.3%-76.0%]. In 95.6% [CI 92.7%-97.4%] of assessments a reading was obtained within three attempts. The median satisfaction score was 95.6 [IQR = 92.2–99.0]. Significantly higher checklist completion rates were observed with single-probe fingertip devices (p<0.001) and children 12–59 months (p<0.001). We found higher satisfaction

**Data Availability Statement:** All data underlying the findings reported in this manuscript is openly available on the data repository figshare (DOI: 10.6084/m9.figshare.21764123).

**Funding:** This work was supported in part, by the Bill & Melinda Gates foundation [OPP1054367 to Malaria Consortium]. Under the grant conditions of the foundation, a Creative commons Attribution 4.0 Generic License has been assigned to the manuscript. The funders had no role in study design, data collection and analysis, decision to publish, or preparation of the manuscript.

**Competing interests:** The authors have declared that no competing interests exist.

scores in South Sudan (p<0.001) and satisfaction varied slightly between devices. From a usability perspective single-probe devices for all age groups should be prioritized for scaled implementation. Further research on easy to use and accurate devices for infants is much needed.

## Introduction

While 75% of countries are estimated to meet the targets of SDG 3.2 "ending preventable deaths of newborns and children under five years of age" by 2030, low-income countries in sub-Saharan Africa and South Asia are lagging [1]. Given that almost half of the world's population of children under five live in these regions [2] acceleration of progress in those areas is desperately needed.

The uptake of the World Health Organization and UNICEF strategies "integrated community case management" (iCCM) and "integrated management of childhood illness" (IMCI) that aim to improve quality of care for child health services in primary care and the community level has been successful [2]. The effect of the programs on neonatal mortality rates and under-five mortality rates has however been questioned [3]. Suboptimal community care for children has been linked to the absence of diagnostic tools to identify children in need for escalated treatment in a community setting. Currently IMCI and iCCM guidelines rely heavily on clinical signs for classification of disease severity [4] that are however difficult to assess correctly for health workers [5]. Clinical signs also lack precision and specificity with regards to common childhood illnesses [6]. These findings indicate a need for more precise tools to identify severely ill children in a community setting.

Next to clinical signs, blood oxygen levels (SpO2) ≤90% are also mentioned as a criterion for referral in the IMCI [4]. Hypoxaemia (SpO2 ≤90%) is a common symptom of neonatal disorders, lower respiratory infections and acute febrile illness indicating a life-threatening severity of the disease that requires timely recognition and treatment [7]. These conditions are among the five most common causes of deaths in children under five [1]. Thus, early detection of hypoxaemia in children in community health settings could help to improve child survival through identifying the severely ill for appropriate treatment and referral.

Pulse oximetry is an established, non-invasive, and effective method to identify and monitor hypoxaemia as an indicator of severe illness in children [6,8,9], but while routinely available in high resource settings, this method is not commonly available in low-income countries [10]. In such settings the introduction of pulse oximetry to health workers with limited skills could, however, increase correct diagnosis and timely recognition of severely ill children [11–13]. Their use in the hands of primary health workers has been associated with higher trust of caregivers [14]. Additionally, pulse oximetry has been deemed highly cost-effective when used in combination with IMCI and could potentially save 148,000 lives in 15 high burden countries [12]. The devices are becoming increasingly affordable [8], and the use of pulse oximeters facilitates a more effective use of oxygen [7] which is highly relevant in the light of the ongoing COVID-19 pandemic.

Following the scientific insights about the effect of pulse oximetry on health outcomes and cost-effectiveness, further investigations into the implementation in primary care and community health settings have been made. The performance of handheld pulse oximeters when used by community health workers (CHWs) and first level health facility workers (FLHFW) in a controlled study setting [15], as well as the operational feasibility of using such devices in

primary care [16] have been proven satisfactory. Additionally, to these assessments the usability of a device is crucial when deciding whether a tool can have a positive impact in the real world [17]. So far, evidence examining usability of pulse oximeters when used by CHWs or FLHFWs in their routine settings is scarce. Only two studies could be identified where a specific probe or device was tested for usability as part of an iterative design process [18,19]. However, no study examining different usability aspects across varying contexts was found. Little is therefore known about challenges frontline health workers might face when using pulse oximeter devices in their routine work setting.

Various multidimensional approaches exist to measure the abstract concept of usability [20–22]. In a human factor-oriented approach defined by Nigel Bevan the usability of a product is evaluated in relation to the implied needs of users within a specified setting [23]. This approach that values context and situational aspects of usability is especially applicable for this study as pulse oximeter usability is to be assessed for a new context of use in a community health setting. In Bevan's approach three components, effectiveness, efficiency, and satisfaction describe the quality of use [23]. They represent the most commonly measured attributes of usability in healthcare and other fields [22,24].

This study aimed to understand the usability of pulse oximeters when used by CHWs and FLHFWs as screening tools for severe illness in children under five in low resource settings through determining effectiveness, efficiency, and satisfaction of five devices in four countries. Through this we generated valuable evidence that informs scaled implementation of pulse oximetry in low-income settings.

## Methods

This cross-sectional study is part of a multi-centre, prospective, two-stage observational study developed to assess performance, acceptability, and usability of pulse oximeter and respiratory rate counter devices used by CHWs and FLHFWs. The protocol for the multi-centre study has been published [25], as well as findings on performance in a controlled hospital setting [15]. This study focuses on the usability of two fingertip and three handheld pulse oximeter devices that have been part of the previous performance study [15]. We have included both fingertip and handheld devices in this analysis, regardless of performance results, to capture a complete picture of pulse oximeter device usability that can inform future device design processes. A detailed description of the devices can be found in Fig 1. Laboratory usability tests evaluate if a specified group is able to use a tool as intended, and field function studies are chosen for effect evaluation in an authentic context [17]. For this study, typical aspects of usability tests and field function studies such as measuring task completion rates and user satisfaction perceptions [17] were transferred to a routine setting.

### Study setting

The study was set in three low-income countries (Ethiopia, South Sudan, and Uganda) and one lower-middle-income country (Cambodia). All four countries were implementing iCCM and IMCI guidelines through their respective Ministry of Health at the time of data collection [25]. The study sites were located in Ratanak Kiri province in Cambodia, the Southern Nations and Nationalities and People's Region (SNNPR) in Ethiopia, Northern Bahr el Ghazal state in South Sudan and in the South-Central Region of Uganda. Altitude levels varied between study sites from elevations <500m in Ratanak Kiri province and Northern Bhar el Ghazal State, 1000-1500m in Mpigi District (South-Central Region of Uganda) and 1500-2500m in the SNNPR in Ethiopia [26]. The project was embedded in CHW's and FLHFW's routine care protocols. FLHFW included in this study work in the lowest level of primary care facilities in

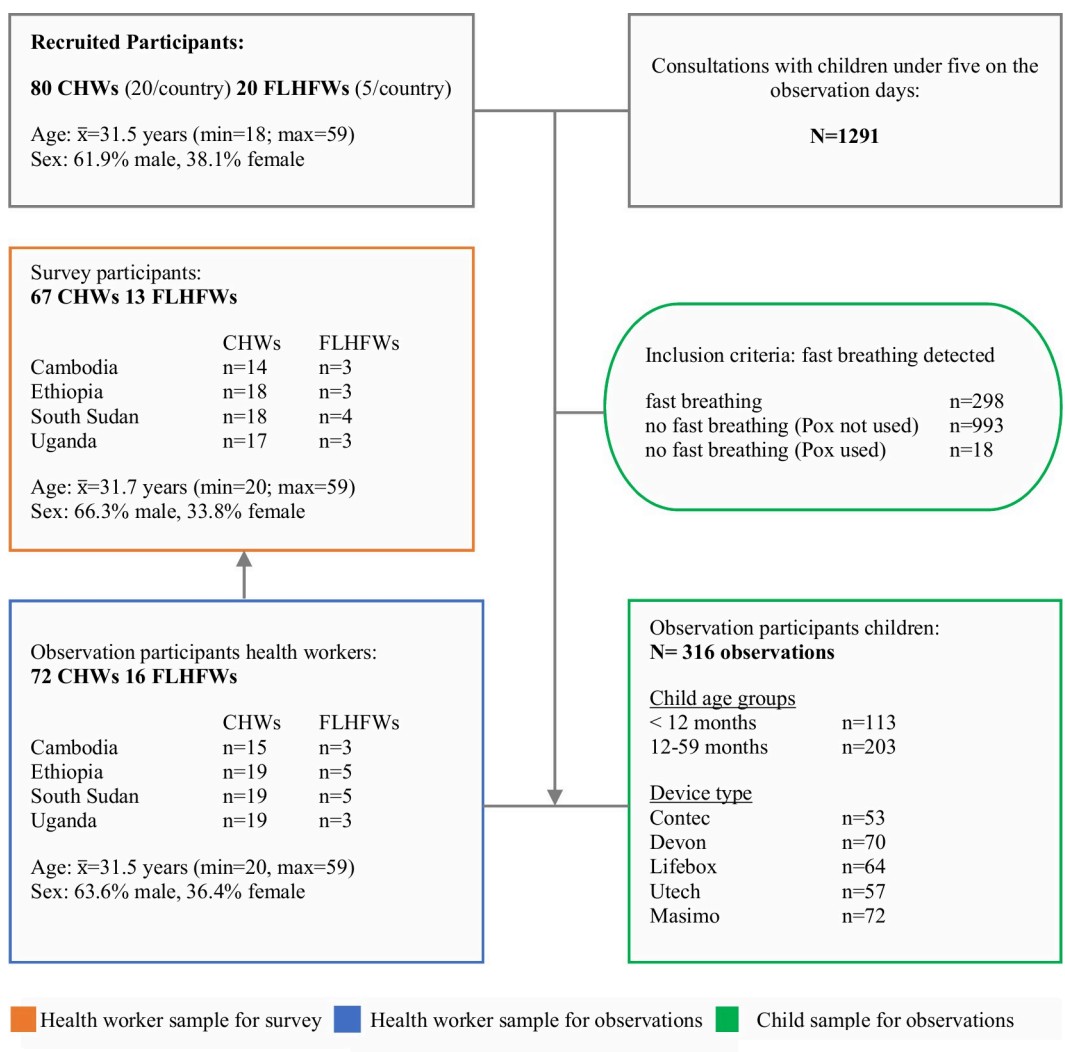

**Fig 1. Pulse oximeter devices used in the study, overview.**

the respective countries, CHW work within the communities receiving patients in their own homes or public spaces of their communities. Further information on health indicators and health system settings in the study sites can be found in the published protocol [25].

## Data collection

Data was collected from February to December 2015. All health workers participating in this study completed a two-day training course on using a pulse oximeter and were assigned one of the devices to use in their daily work. Prior to data collection community sensitization was organized to increase awareness of improved diagnostic tools in the community to ensure patient engagement and support study enrolment. Data collectors used pilot tested paper-based forms. The health workers were visited by data collectors three times over the course of three months and per visit a maximum of five consultations of children under five were observed. During one visit an orally administered survey that assessed perceived usability and attitude towards pulse oximeter devices was conducted. Ultimately the data was entered into a

digital format using EpiData version 3.1 [27] ensuring anonymity and following a data entry protocol.

## Participants

CHWs and FLHFWs who participated in the study on performance in a controlled setting and whose workplace was within 20km reach of a health facility with oxygen equipment were informed about the data collection for this study and asked for their willingness to participate. Children under five presenting with fast breathing on the day a researcher observed the health worker were sampled based on convenience. This inclusion criteria had been set based on an interpretation of IMCI guidelines [4]. Ethical principles were respected for the recruitment of all participants and only those that provided written informed consent were included. In the case of children, caretakers provided informed consent. Initially 100 health workers (80 CHWs, 20 FLHFWs) were recruited for the study but since not all of them assessed a child that fit the inclusion criteria, on the day of field observations, the final number of participants was 72CHWs and 16 FLHFWs. In total 298 children under five with confirmed fast breathing were assessed with a pulse oximeter and 18 additional observations were recorded on children without fast breathing. Since this analysis focused on device usability and not clinical outcomes all observations were included, resulting in a sample of 316 pulse oximetry assessments. The survey with questions on perceived usability and attitude towards the devices was conducted with 80 participants sampled from the observation participants group based on convenience. A detailed profile of the recruitment process can be found in Fig 2.

## Sample size

The sample size of this study was limited by the conditions of participation in the study on performance in a controlled setting and the availability of an oxygen providing facility within 20km reach. While a priori sample size calculations were conducted for the performance trial, they were not possible for the usability aspect of the study. However, a difference in agreement with a reference standard of 0.22 had been found between device groups in the study on pulse oximeter performance [15]. The necessary sample size to detect a similar difference in this study, given a significance level of $\alpha = 0.05$ and a power $(1-\beta) = 0.8$ would be N = 172, that is N = 86 observations in each group. The available sample for testing for difference between the two groups fingertip and handheld devices in this study was N = 244 with N = 134 observations in group one and N = 110 observations in group two.

## Outcomes

We measured three aspects of usability: effectiveness, efficiency, and satisfaction. Effectiveness was defined as full adherence to a binary coded observational checklist with nine items. The items represent successive steps within the process of acquiring a correct blood oxygen measurement with a pulse oximeter. The items one, two, three, eight, and nine in the checklist are applicable for all devices, fingertip and handheld, items four, five, six, and seven however are only applicable for handheld devices as they relate to probe choice and connection. Fingertip devices are single-probe devices. Other items of the checklist data collection form were excluded either because they were not applicable for pulse oximeter measurements or due to inconclusive coding and missing data. The checklist had been developed by Malaria Consortium in collaboration with pneumonia and paediatrics experts. The binary coding was defined by whether a step was completed correctly or not. A binary coded variable, *failure*, was used to measure efficiency. This variable indicates if a pulse oximeter measurement could be obtained within three attempts or not. Efficiency is typically a time-controlled measure but can also be

**Contec fingertip pulse oximeter (Model: CMS50QB)**
Probe is attached to patient's finger or toe for measuring Sp02 and heart rate. The device is supplied with two rechargeable batteries and is recommended for paediatric patients. CE approval [a] as a class IIb [b] medical device. Price ca. 20$.

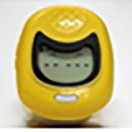

**Devon fingertip pulse oximeter (Model:PC600)**
Probe is attached to patient's finger or toe for measuring SpO2 and heart rate. The device has a rechargeable battery and is recommended for paediatric patients. CE approval [a] as a class IIb [b] medical device. Price ca. 50$.

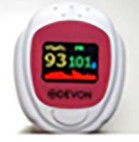

**Lifebox handheld pulse oximeter (Model: AH-M1)**
Handheld device with reusable adult, neonatal and paediatric probes for measuring Sp02 and heart rate. Includes visual and audible alarms. It is both battery and mains powered and equipped with a rechargeable lithium battery. Inlcudes two-year warranty. CE approval [a] as class IIb [b] medical device. Price ca. 250$.

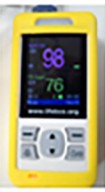

**Utech handheld pulse oximeter (Model: UT100)**
Handheld device with adult reusable probe as standard for measuring Sp02 and heart rate. Neonatal and paediatric probes available. Rechargeable batteries need to be purchased additionally. Includes one-year warranty. CE approval [a] as class IIb [b] medical device. Price ca. 100$.

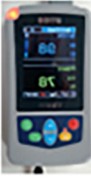

**Masimo mobile phone pulse oximeter (Model: iSpO2 Rx)**
Handheld device operating on Android smartphones and iPhones. Single and multiuse adult, neonatal and paediatric probes for measuring SpO2 and heart rate available. The device features low perfusion and motion software. CE approval [a] as class IIb [b] medical device. Price ca. 250$ (plus costs of mobile phone)

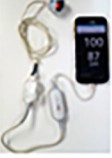

**Fig 2. Flow chart participant recruitment.** [a] Confirmation that a medical device meets the safety, health and environmental protection requirements of the European Economic Area (EEA) [28]. [b] devices intended for monitoring possibly dangerous changes in vital parameters [29].

defined through a fixed amount of tries for example as a failure rate [22]. An orally administered questionnaire was used to measure health worker satisfaction. It consisted of 19 items rated on a five-point Likert-scale ranging from 1 = not at all to 5 = very. All 19 items were included as separate ordinal variables. Furthermore, explanatory variables, containing demographic and contextual information, were used in the analysis. These were: anonymous identification code of the health worker, cadre, country, child age, and device.

## Statistical analyses

Statistical analyses were conducted using StataSE 17 [30]. Listwise deletion was performed for observations that showed missing values and a full case analysis was carried out. To describe effectiveness and efficiency, frequencies and proportions were calculated with binary data [31]. Proportions of assessments with full adherence and non-adherence to the observational checklist were calculated. Additionally, task completion rates were calculated per step of the

observational checklist. To describe satisfaction, a summated scale was formed with all 19 Likert items of the questionnaire. The score was subsequently standardized to a value between 20 and 100. Thus, assuming someone had rated all 19 Likert items with the lowest possible score of 1, this would pertain to a summated standardized score of 20 ((19*1*100)/(19*5)). The summated standardized scores are expressed in the continuous variable satisfaction score for every survey participant. For this study the survey data is treated as interval data based on the principle that if multiple Likert items are combined into a Likert Scale that describes a more abstract concept such as satisfaction, the combined score can be treated as numerical [32]. To determine internal consistency of the scale Cronbach's alpha was used [32,33]. Values between 0.7 and 0.9 are known to confirm reliability [32,33]. For this study Cronbach's alpha for all included survey items was $\alpha = 0.86$. Due to the non-normal distribution of the continuous outcome variable satisfaction score, confirmed through Skewness and Kurtosis as well as Shapiro Wilk test, the median and interquartile ranges were used to describe the outcome. Confidence intervals and standard errors were calculated to indicate the precision with which the estimates approach the true population value [31]. To explore influencing factors on the two main outcomes effectiveness and satisfaction hypothesis tests for differences between independent groups were conducted. We used $Chi^2$ tests for binary data, Mann-Whitney-U tests for non-parametric continuous data and Kruskall-Wallis tests for differences between multiple groups. All hypothesis tests were two-sided and significance levels were set at $\alpha = 0.05$. Random effects models built with the variables used in our analysis did not show evidence of a panel effect in the data.

## Reflexivity statement

The multi-centre observational study that data for this study is derived from was designed by Malaria Consortium researchers in the respective countries in collaboration with the research team in London. Thus, local research and policy priorities were reflected in this research. All researchers responsible for data collection activities in the countries are listed as co-authors in this publication or acknowledged at the end of the text if they did not fulfil authorship criteria. The main author of this publication has not been part of the data collection process. To validate the study's aim and analysis plan, a meeting with researchers from the respective countries was conducted. Videos that were used as part of data collection helped the main author to familiarize herself with the context. Only communities in 20kms reach of an oxygen providing health facility were included in the study to ensure safety and the possibility of treatment for all study participants.

## Ethics

Ethical approval for this study was issued through the SNNPR Health Bureau Health Research Review Committee (Ref: 6-19/10342), the Uganda National Council for Science and Technology (Ref: HS 1585), the Research and Ethics Committee at the Government/Ministry of Health South Sudan (no reference number provided), the National Committee for Health Research Cambodia (Ref: 0146NECHR) and the Regional Ethics Committee Stockholm Sweden (Ref:2017/4:10). Participants were included in the study after written informed consent had been acquired and for data entry the health worker's names were replaced by an identification code.

## Results

In total we included 316 observations of pulse oximeter assessments on children under five by 72 CHWs and 16 FLHFWs and 80 completed questionnaires administered on 67 CHWs and

## Effectiveness

72 observations were excluded from effectiveness analysis due to missing data (22.8%). Thus, the sample for this analysis consisted of 244 observations by 60 CHWs and 14 FLHFWs. A summary of baseline characteristics of health worker participants and children sample as well as the distribution of missing data across checklist steps can be found in the (S1 and S2 Tables). The findings show a checklist completion rate of 78.3% [95% CI 72.6–83.0]. Subgroup analyses showed a significant difference in checklist completion rates between fingertip (95.5%, 95%CI 89.5–98.1) and handheld devices (64.2%, 95% CI 55.7–71.9), p<0.001, and between infants <12 months (55.3%, 95% CI 44.6–65.5) and children 12–59 months (90.6%, 95% CI 84.9–94.3), p<0.001. No difference was found between cadres of health workers or countries (Table 1). Task completion rates of single checklist items can be found in Table 2. Step 4 which involved choosing the correct probe for the child's age showed the lowest task completion rate (68.7%, 95% CI 60.3–76.0). Task completion rate of Step 4 was lower among children <12 months (28.3%, 95% CI 17.1–42.9) than children 12–59 months (89.8%, 95% CI 81.4–94.6). Task completion rates of checklist items by age group can be found in the (S3 Table). Among handheld devices task completion rates of Step 4 ranged between 64.9% and 78.4%.

## Efficiency

For efficiency measurements all observations (N = 316) could be included in the analysis, characteristics of participants can be found in Fig 2. In 95.6% of observations health workers obtained a pulse oximeter reading within three attempts. Failure rates were below 10% for all devices. We observed the highest failure rate at 8.8% for the handheld device from U-Tech. We did not find a significant difference in the efficiency measure between the two device groups, p = 0.416. All findings of our efficiency analysis can be found in Table 3.

## Satisfaction

The satisfaction analysis comprises 80 health workers, 67 CHWs and 13 FLHFWs, participant characteristics can be found in Fig 2. The median satisfaction score was 95.6 (IQR = 92.2–98.9,

**Table 1. Effectiveness outcome: Frequency of full task completion, task completion rates with 95% confidence intervals and p-values.**

|  | N | Freq. | Task completion rate (%) | 95% CI | | p-value |
|---|---|---|---|---|---|---|
| **Overall** | 244 | 191 | 78.3 | 72.6 | 83.0 | |
| **Device groups** | | | | | | <0.001 |
| Fingertip devices | 110 | 105 | 95.5 | 89.5 | 98.1 | |
| Handheld devices | 134 | 86 | 64.2 | 55.7 | 71.9 | |
| **Cadre** | | | | | | 0.278 |
| CHW | 201 | 160 | 79.6 | 73.4 | 84.6 | |
| FLHFW | 43 | 31 | 72.1 | 56.9 | 83.5 | |
| **Child age** | | | | | | <0.001 |
| < 12months | 85 | 47 | 55.3 | 44.6 | 65.5 | |
| 12–59 months | 159 | 144 | 90.6 | 84.9 | 94.3 | |
| **Country** | | | | | | 0.900 |
| Cambodia | 34 | 26 | 76.5 | 59.4 | 87.8 | |
| Ethiopia | 93 | 71 | 76.3 | 66.6 | 83.9 | |
| South Sudan | 93 | 75 | 80.7 | 71.3 | 87.5 | |
| Uganda | 24 | 19 | 79.2 | 58.5 | 91.1 | |

**Table 2. Frequency of task completion and task completion rates per checklist item.**

|  | N | Freq. | Task completion rate (%) | Std. error | 95% CI | |
|---|---|---|---|---|---|---|
| *Step1*<br>*"Child is calm and sitting in a reclined position"* | 244 | 237 | 97.1 | 1.1 | 94.1 | 98.6 |
| *Step2*<br>*"The child's upper or lower limb digits are exposed"* | 244 | 244 | 100.00 | - | - | - |
| *Step3*<br>*"Health worker turns device on correctly"* | 244 | 244 | 100.00 | - | - | - |
| *Step4* (handheld devices only)<br>*"Health worker uses the appropriate probe for the child's age"* | 134 | 92 | 68.7 | 4.0 | 60.3 | 76.0 |
| *Step5* (handheld devices only)<br>*"Health worker checks and cleans probe sensor"* | 134 | 132 | 98.5 | 1.1 | 94.2 | 99.6 |
| *Step6* (handheld devices only)<br>*"Health worker connects probe sensor to unit"* | 134 | 134 | 100.00 | - | - | - |
| *Step7* (handheld devices only)<br>*"Health worker attaches probe sensor correctly"* | 134 | 134 | 100.00 | - | - | - |
| *Step8*<br>*"Health worker correctly distinguishes between Sp02 and pulse reading on display"* | 244 | 244 | 100.00 | - | - | - |
| *Step9*<br>*"Health worker records two readings"* | 244 | 239 | 98.0 | 0.9 | 95.2 | 99.2 |

min = 74.4, max = 100) indicating an overall high satisfaction with the pulse oximeters among health workers. The highest median satisfaction score was observed in South Sudan (MED = 100, IQR = 97.7–100) and a Kruskal-Wallis Rank Test showed a statistically significant difference between countries, p<0.001. The findings show no significant difference in satisfaction between device types, p = 0.965, or health worker cadres, p = 0.819 (Table 4).

## Discussion

The findings showed that while CHWs and FLHFWs in Cambodia, Ethiopia, South Sudan, and Uganda can use fingertip pulse oximeters very effectively within their routine work, the usability of handheld devices was significantly lower. Among checklist steps, choosing the correct probe for the child's age in multiprobe-handheld devices proved to be the most difficult task. The findings also revealed low usability of pulse oximeters in infants under one. Health workers reported high satisfaction with the additional diagnostic tools. Failures to obtain a reading were rare, indicating high efficiency of use.

Previous investigations on usability of pulse oximeters have established effective use through time-controlled measures [16,18,34]. Task completion rates, as used in our study, provide a meaningful addition to the established time-controlled measurements, especially for

**Table 3. Efficiency outcome: Frequencies and percentages of successfull measurements.**

|  | Percent | Frequency | Std. error | 95% CI | |
|---|---|---|---|---|---|
| Reading obtained | 95.6 | 302 | 1.2 | 92.7 | 97.4 |
| Failure to obtain reading | 4.4 | 14 | 1.2 | 2.7 | 7.4 |
| **Failure rates per device** |  |  |  |  |  |
| Contec | 1.9 | 1 | 1.9 | 0.3 | 12.3 |
| Devon | 4.3 | 3 | 2.4 | 1.4 | 12.5 |
| Lifebox | 3.1 | 2 | 2.2 | 0.8 | 11.7 |
| Utech | 8.8 | 5 | 3.8 | 3.7 | 19.5 |
| Masimo | 4.2 | 3 | 2.4 | 1.3 | 12.2 |

**Table 4. Five-number summary of satisfaction scores overall and per device group, country, and cadre.**

|  | Median | IQR | Min-Max | 95% CI | | z-value | p-value |
|---|---|---|---|---|---|---|---|
| **Overall** | 95.6 | 92.2–98.9 | 74.4–100 | 94.1 | 97.7 | | |
| **Country** | | | | | | | < .001 |
| Cambodia | 93.3 | 92.2–95.3 | 77.7–98.9 | 92.2 | 95.3 | | |
| Ethiopia | 96.8 | 85.3–99.0 | 82.1–100 | 88.4 | 98.9 | | |
| South Sudan | 100 | 97.7–100 | 88.2–100 | 97.7 | 100 | | |
| Uganda | 94.7 | 89.7–96.8 | 74.4–100 | 90.8 | 96.6 | | |
| **Device groups** | | | | | | 0.044 | .503 |
| Fingertip devices | 95.9 | 92.9–98.8 | 84.4–100 | 93.3 | 97.8 | | |
| Handheld devices | 95.6 | 90.6–99.5 | 74.4–100 | 93.3 | 98.9 | | |
| **Cadre** | | | | | | 0.229 | .480 |
| CHW | 95.6 | 92.2–99.0 | 74.4–100 | 94.1 | 97.6 | | |
| FLHFW | 98.8 | 85.6–99.0 | 77.8–100 | 84.9 | 99.6 | | |

community settings where health workers value accuracy over time [14]. Taking into account the risks involved with task failure [35], full checklist adherence during pulse oximeter measurements in community settings seems justified. Overall CHWs and FLHFWs included in this study adhered to all checklist steps in 78.3% of cases which can be considered a satisfactory task completion rate but leaves room for improvement [35]. Our analyses focusing on device types and age groups revealed areas of lower usability where pulse oximeters need to be improved and adapted to better fit the context of low resource community health contexts.

The sensitivity analysis of task completion rates per checklist item showed that choosing the correct probe for a child's age presented a challenge to CHWs and FLHWs. Results in this aspect can be considered unsatisfactory [35]. It is known that the fit of a probe for the child's height and weight is difficult but essential for correct measurements especially for young infants and neonates [6,7]. Our findings additionally suggest that adaptions in the measurement procedure between different child ages are complicated and decrease usability. Therefore, attempts of designing a universal paediatric probe, like the Lifebox LB-01 probe [34] should be supported and prioritized. This universal probe combined with a handheld box has been shown to be equivalent to the use of separate neonate wrap and paediatric clip probes in performance [34].

The observed difficulties with probe choice also delineate the difference in effective use between fingertip and handheld devices established in this study. Because fingertip devices only provide one clip for all ages, Step 4 in the checklist was not applicable for these observations. Taking into account very high task completion rates in all other steps, the impact of probe choice on observed differences between the two groups can be considered high. Fingertip devices have previously been found to perform poorly when used by CHWs regarding the overall agreement of SpO2 measurements with a reference standard [15]. This contrast between better performance of handheld devices and higher usability of fingertip devices underlines the need for a handheld device with a universal probe. Such devices combine the good performance of handheld units with high usability of single probe pulse oximeter devices found in this study. Both accuracy as well as high usability are very important in lower levels of low-resource health care settings in order to precisely and efficiently refer severely ill children to higher levels of care. Our findings on usability, together with previous findings on accurate performance [15] support the scaled use of such pulse oximeters in low resource community health contexts.

Besides the device type we also found a difference in the effectiveness measures between infants <12 months and children aged 12–59 months. Most previous research indicates that pulse oximeters are more difficult to use in younger children [6,14,18], except for one study in Pakistan that reported no influence of child age on usability [16]. This study used different inclusion criteria and a time-controlled measurement that did not include the time taken to choose the correct probe for a child's age [16] which is likely to have led to different estimations. Infants constitute an important target group for pulse oximetry due to significantly higher prevalence of hypoxaemia in this age group [36] and the special role of screening for congenital heart disease in neonates [37]. Besides these important clinical factors, the special need for appropriate care in infants and neonates should also be seen in the global perspective of a rising proportion of deaths of children under five [1]. While the tests performed in this study do not allow for concrete conclusions to be drawn on a correlation between the effect of device type and child age on effectiveness, the findings show lower task completion rates in infants of the most difficult step "choosing the correct probe for the child's age" which affects effectiveness results of handheld devices. This further supports the use of single-probe devices, also for the highly important age group of children under one. However, further research on difficulties in use for this age group is necessary to inform the design of a universal probe for infants and children above one year alike.

Our results of very high satisfaction with the routine use of pulse oximeters are in line with previous research in low and high resource settings [19,38]. Compared with benchmarks of the standardized System-Usability-Scale satisfaction results across all four countries of this study can be deemed very high [39]. A very small positive difference in satisfaction scores of South Sudanese health workers might be related to low clinical training and limited diagnostic resources which might have positively influenced their attitude towards pulse oximeters [14].

It is also important to discuss our results with regard to the context of use as it is an important factor for evaluating usability [23]. Since we did not find any difference in effectiveness between health worker cadres, it is likely that CHWs, as well as FLHFWs have the necessary skills to operate pulse oximeters. This is also supported by the lack of difference between countries, where training and education vary. The consistency of our usability results across countries and educational backgrounds suggests that our findings could be transferable across low resource settings. However, the use of pulse oximeters is also strongly linked to further organizational factors and should be considered with relation to referral recommendations and correct disease classifications [16]. These implications are especially important for further implementation considerations because the effect of pulse oximetry in community health and primary care settings on childhood survival is strongly linked to functioning referral systems and the provision of oxygen in higher level health facilities [12].

## Methodological considerations, strengths, and limitations

The main strength of this work is its integration within a routine care setting, observing participants within their real-life environment, which enables conclusions to be drawn for the actual work context. The inclusion of countries from four different geographic locations and two different income levels added towards transferability. Another factor that strengthens the validity of our findings is the use of three different aspects (two objective and one subjective) to describe usability of pulse oximeters, thus contributing to a deeper understanding of the abstract concept of usability [23].

There are several limitations that need to be mentioned. Although the observational nature is a key strength of the study, the Hawthorne effect, as an inherent limitation to observational designs, could have introduced bias to the observational findings. Social desirability bias could

be present in both observational and survey findings. Regarding the data collection tools, the observational checklist used for the assessment of effectiveness only captured task completion versus failure to complete a task. Hence, the quality of completion, as well as partial task completion could not be taken into consideration, even though this would have provided a more detailed picture of effectiveness [22]. The pulse oximeter waveform that provides an estimate of the intra-arterial waveform is not displayed on all devices in this study and its observation was not included in the checklist tasks. This can be seen as a limitation because the process of matching results with the waveform to increase accuracy has been discussed as a complicated process [40]. Including the observation of the waveform in the checklist thus might have resulted in lower task completion rates.

Furthermore, statistical limitations are noteworthy. Firstly, a full case analysis was conducted for the efficiency analysis, even though the data was not missing completely at random. However, listwise deletion was preferred over multiple imputation given the considerable size and pattern of missing data which suggested problems during the data collection process, especially in Uganda where most of the missing data was concentrated (S1 Table). We did not see a link between health worker's characteristics and missing data which supports a full case analysis [41]. While pairwise deletion was not possible for the outcome of checklist completion, task completion rates per checklist item were also calculated using pairwise deletion, which yielded similar results as listwise deletion and did not impact the conclusions drawn from the findings. Results of this sensitivity analysis can be found in the supplementary material (S4 Table). Second, the questionnaire used to estimate satisfaction was not standardized. While internal consistency was confirmed through Cronbach's alpha we did not check for uni-dimensionality through factor analysis. Third, there are some limitations concerning sample size; while the sample (N = 244) was large enough to detect a difference between two groups similar to the differences detected in a previous performance study [15], the sample size for the satisfaction component was considerably smaller (N = 80) and therefor hypothesis tests for this aspect provide only limited informative value.

## Conclusion

Our findings on usability of pulse oximeter devices, together with previous research on their performance [15] and the effect of pulse oximetry on health outcomes [6,11,12] provide evidence supporting the scaled use of single probe handheld devices at lower health care levels in low resource settings. However, further research focusing on uncomplicated and appropriate use in infants is much needed, due to the usability challenge for this age group identified in our study.

## Supporting information

**S1 Checklist. STROBE statement—checklist of items that should be included in reports of *cross-sectional studies*.**
(DOC)

**S1 Table. Descriptive parameters of study participants.**
(XLSX)

**S2 Table. Description of checklist variables.**
(XLSX)

**S3 Table. Frequency of task completion and task completion rates per checklist item by age group.**
(XLSX)

**S4 Table. Frequency of task completion and task completion rates per checklist item, pairwise deletion.**
(XLSX)

## Acknowledgments

We want to thank all Malaria Consortium country research teams who assisted in the development and implementation of this research, Ministry of health staff in all countries, and health workers, caregivers and children who participated in this study for their dedication, time, and contributions to this project. Additionally, our special thanks shall be expressed to our former colleague Dr. Morris Okwir, for his contributions to the conceptualization, methodology, investigation and project administration of this study in South Sudan.

## Author Contributions

**Conceptualization:** Theresa Pfurtscheller, Kevin Baker, Tedila Habte, Kévin Lasmi, Lena Matata, Akasiima Mucunguzi, Jill Nicholson, Anthony Nuwa, Max Petzold, Mónica Posada González, Anteneh Sebsibe, Tobias Alfvén, Karin Källander.

**Data curation:** Theresa Pfurtscheller.

**Formal analysis:** Theresa Pfurtscheller.

**Funding acquisition:** Kevin Baker, Karin Källander.

**Investigation:** Tedila Habte, Lena Matata, Akasiima Mucunguzi, Anthony Nuwa, Mónica Posada González, Anteneh Sebsibe.

**Methodology:** Theresa Pfurtscheller, Kevin Baker, Tedila Habte, Kévin Lasmi, Lena Matata, Akasiima Mucunguzi, Jill Nicholson, Anthony Nuwa, Max Petzold, Mónica Posada González, Anteneh Sebsibe, Tobias Alfvén, Karin Källander.

**Project administration:** Kevin Baker, Tedila Habte, Lena Matata, Akasiima Mucunguzi, Jill Nicholson, Anthony Nuwa, Mónica Posada González, Anteneh Sebsibe, Karin Källander.

**Supervision:** Kevin Baker, Jill Nicholson, Tobias Alfvén, Karin Källander.

**Validation:** Max Petzold.

**Visualization:** Theresa Pfurtscheller.

**Writing – original draft:** Theresa Pfurtscheller.

**Writing – review & editing:** Kevin Baker, Tedila Habte, Kévin Lasmi, Lena Matata, Akasiima Mucunguzi, Jill Nicholson, Anthony Nuwa, Max Petzold, Mónica Posada González, Anteneh Sebsibe, Tobias Alfvén, Karin Källander.

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
