## [Decision Letter · Decision Letter 0]

31 Jan 2023

PGPH-D-22-02104

Usability of pulse oximeters used by community health and primary care workers as screening tools for severe illness in children under five in low resource settings

A cross-sectional study in Cambodia, Ethiopia, South Sudan, and Uganda

Dear Dr. Pfurtscheller,

Thank you for submitting your manuscript to PLOS Global Public Health. After careful consideration, we feel that it has merit but does not fully meet PLOS Global Public Health’s publication criteria as it currently stands. Therefore, we invite you to submit a revised version of the manuscript that addresses the points raised during the review process.

We look forward to receiving your revised manuscript.

Kind regards,

Sakib Burza, MBChB, MRCGP, MSc, PhD

Academic Editor

Journal Requirements:

1. In the online submission form, you indicated that your data will be submitted to a repository upon acceptance.  We strongly recommend all authors deposit their data before acceptance, as the process can be lengthy and hold up publication timelines. Please note that, though access restrictions are acceptable now, your entire data will need to be made freely accessible if your manuscript is accepted for publication. This policy applies to all data except where public deposition would breach compliance with the protocol approved by your research ethics board. If you are unable to adhere to our open data policy, please kindly revise your statement to explain your reasoning and we will seek the editor's input on an exemption. Please be assured that, once you have provided your new statement, the assessment of your exemption will not hold up the peer review process.

Additional Editor Comments (if provided):

Dear Authors,

Many thanks for submitting this very interesting paper. The overall quality of the research (and pertinence of the paper) is very good, but there are areas of clarification and improvement that will make a substantial impact on both its readability and impact. We very much look forward to receiving a revised version of this important manuscript.

Sakib

Reviewers' comments:

Reviewer's Responses to Questions

**Comments to the Author**

1. Does this manuscript meet PLOS Global Public Health’s publication criteria? Is the manuscript technically sound, and do the data support the conclusions? The manuscript must describe methodologically and ethically rigorous research with conclusions that are appropriately drawn based on the data presented.

Reviewer #1: Yes

Reviewer #2: Yes

Reviewer #3: Yes

Reviewer #4: Partly

2. Has the statistical analysis been performed appropriately and rigorously?

Reviewer #1: Yes

Reviewer #2: Yes

Reviewer #3: Yes

Reviewer #4: No

3. Have the authors made all data underlying the findings in their manuscript fully available (please refer to the Data Availability Statement at the start of the manuscript PDF file)?

Reviewer #1: No

Reviewer #2: Yes

Reviewer #3: Yes

Reviewer #4: No

4. Is the manuscript presented in an intelligible fashion and written in standard English?

Reviewer #1: Yes

Reviewer #2: Yes

Reviewer #3: No

Reviewer #4: Yes

5. Review Comments to the Author

Reviewer #1: Thank you for asking me to review this interesting manuscript. The authors report the results of an observational cross-sectional study evaluating the usability of pulse oximeters in community settings across South Sudan, Ethiopia, Uganda, and Cambodia. They find that pulse oximetry can be performed effectively, efficiently, and healthcare workers derive high levels of satisfaction from using the devices. Simpler finger-tip devices appear to be more effective but this must be contextualised with inferior performance reported in previous studies.

The manuscript is well written and a pleasure to read. It reports important findings. I congratulate the authors on their work. My comments have been prepared according to the STROBE guidelines for cross-sectional studies. I would encourage the authors to complete this reporting checklist and submit it with the revised version of their manuscript.

Title / Abstract

No comments. Well written – clear and concise.

Introduction

No major comments. Well written – background presented and objectives stated.

- Line 65: Minor comment – perhaps “identifying” might be better than “filtering out”

- Lines 71: Minor comment – think should be “in the hands” rather than “at the hand”

Methods

Study design, setting, participants, exposure variables, outcomes, and data collection clearly outlined.

- Please explain / justify why listwise rather than pairwise deletion was used to handle missing data?

Results

- Why were observations with missing data excluded from the task completion rates per checklist item? This relates to the choice of listwise rather than pairwise deletion. Would it not have been better to use pairwise deletion to reduce the proportion of missing data?

- Please provide numerators and denominators for task completion rate percentages provided in Tables 2 and 3.

- Please clarify which steps pertain to which devices – I think Step 4 only relates to the handheld devices but it is not clear whether this is also the case for Steps 5-7? I note this is expanded upon in the Discussion but would be useful to present this earlier.

- Lines 232/233: I cannot see where the statement “overlapping confidence intervals between the handheld devices indicated no significant difference of task completion of Step 4 between specific devices (Table 3)” is supported in Table 3? Suggest to remove the reference to Table 3 and replace with “data not shown” if this is indeed the case.

- It would be helpful to report the refusal rate – both of HCWs and caretakers.

Discussion

- Lines 259/260: I am not sure that from the results presented it can be said that satisfaction varied between devices (95.9 vs. 95.6; p = 0.503)? Please review this and provide supporting data if true.

- Line 284: Given that one of the main findings of this study is the differential effectiveness of handheld vs. finger-tip devices it would be useful to briefly summarise the important aspects of the better performance of handheld devices in this publication (rather than refer the readers to the previous publication). This would help readers contextualise the findings and implications of this study better.

- Line 285: What underlies the differential effectiveness rates in infants and children? Presumably it is that the wrong probe was more often selected for infants? i.e. the root cause of the differential effectiveness in the hand-held and finger-tip devices is the same as the differential effectiveness in the different age groups? If so, this should be explicitly stated to make it clear there is not another factor at play. Alternatively, if there are other possible explanations, this should be expanded upon.

- Lines 333-335: It is not clear why the variability in SpO2 cut-offs for referral according to altitude are mentioned here. They do not seem to play a role in this study?

- Line 268: Minor comment – standardise number of decimal places reported.

Reviewer #2: This is an important work that focuses on one of leading cause of death in children, particularly in low and middle income countries. The authors used sound and appropriate methods to address the research question. The manuscript is well written and clear. The paper deserves to be published provided that the followings comments are addressed.

Reviewer #3: This is a peer review of a manuscript titled "Usability of pulse oximeters used by community health and primary care workers as screening tools for severe illness in children under five in low resource settings A cross-sectional study in Cambodia, Ethiopia, South Sudan, and Uganda" by Pfurtscheller et al.

General comments:

1) This analysis and paper have useful findings and it deserves publication. However, the paper would benefit greatly by further editing for clarity and conciseness throughout prior to full acceptance. And a hard look at the messaging of the paper to bring out important nuance that I feel is currently lacking.

2) While the authors do message the paper's findings in a positive light, my interpretation is perhaps more concerned due to the fact that providers struggled completing tasks on <12 month olds, selecting the correct probe for age, and favoring the most inaccurate probes (finger tip probes). The authors do point these areas out, but I do think they deserve more attention and emphasis. As the authors do correctly state in their discussion section that <12 month olds are the key age range as they represent where the burden of disease and mortality lies, I don't feel like the writing properly emphasizes this issue as really the major finding. On top of this healthcare workers also struggled in selecting the correct probe for age, and favored finger tip proves, which would suggest that measurements may be inaccurate. While the authors do point out these concerns, and do recommend a universal probe/device for all ages, I don't personally interpret the findings as positively as the authors seem to be messaging. I would recommend relooking at this messaging and considering adjusting the tone. Personally the paper really could be redone to message as a stronger call to action on device development for <12 month olds in LMICs. (in reality the message could be: usability is favorable on the lowest risk children (>12 months old) with healthcare workers using the most inaccurate devices (finger tip devices)!!, while usability was UNFAVORABLE on the age range that really matters (<12 month olds, which is a huge problem and we need to urgently address this)

3) Allowing for 3 measurements is generous. Is there any data on <3 measurements? I highly doubt during unobserved care situations providers will be trying 3 times on patients as a general rule.

4) The Hawthorne effect issue is an important one, and the authors do rightly bring this out as a limitation. My concern is that this study observed providers for a few patients, and likely those providers took their time and made sure that all of the patients they saw with the observer had oxygen saturation measurements done. My concern is that when they are not being observed, providers may be more likely to try once on a child, and then move on, especially if patient volumes are high. Or skip the measurement altogether. This does impact the authors main conclusion that "usability is favorable." (as above, in reality the message could be: usability is favorable on the lowest risk children (>12 months old) with healthcare workers using the most inaccurate devices (finger tip devices)!!) The paper would benefit from more insight on the use of the devices outside of this specific study, perhaps brought in or referenced from other studies done on this project. What would be particularly important to understand is what % of patients eligible for measurements had measurements done overall (not just on days of observation)? How representative are the days of observation as compared to unobserved days? This would also help the paper.

Reviewer #4: Thank you for this interesting and useful paper which is overall well done and very well written (the introduction and discussion are particularly good). My main concerns are on the methods and statistics, and in some places consequently, the results, including what is not presented, and their interpretation. My comments are in order from the beginning to the end of the paper as I went through it:

1. Abstract – last sentence: “From a usability perspective single probe devices for all age groups should be prioritized for scaled implementation.” – please clarify – you only mention fingertip and handheld devices in the abstract – which ones are you referring to here when you say “single probe devices”.

2. Methods – outcomes, last sentence: “In accordance with the available data the main outcomes were defined as effectiveness and satisfaction” – please clarify – why was the third outcome (efficiency) not also a “main outcome”? Also, what do you mean by “main outcome”?

3. Statistical analysis: Please can you state what proportion of cases had missing data? (and were consequently excluded). Was missing data concentrated in a few variables only? Was missing data actually quite rare in some analyses as per footnotes in Table S1? Did you consider multiple imputation of missing data? I see in the limitations section of the discussion you write “Firstly, a full case analysis was conducted, even though the data was not missing completely at random. However, multiple imputation as an alternative method would have likely resulted in a stronger bias, given the sample size and considerable proportion of missing data.” Please could you substantiate this with details of the data that was missing (observations and variables within observations – would be good to include this as a supplementary table, or in the main tables) and a reference on when to do and when not to do multiple imputation.

4. Statistical analysis: When you converted the 1-5 Likert scale scores to a numerical 0-100% score did you assume a score of 1 (the lowest) corresponded to zero on the numerical scale, such that answering all 1s would mean 0% on the 0%-100% scale? The min score in Table 4 of your results is 74.4 which would suggest an average of 4 for all questions on the 1-5 Likert scale, which seems very high for the minimum (average of 1(lowest)=0%, 2=25%, 3=50%, 4=75%, 5(highest)=100%)?

5. Statistical analysis: Did you consider regression analyses to look at associations between your outcomes and variables of interest, including multiple variables at once?

6. Results – Table S1 suggests missing data for your effectiveness analysis was concentrated in Uganda, which only had 9 CHW and 1 FLHFW. Please comment on why this happened, and how it might affect interpretation of your results for Uganda.

7. Results, bottom of page 10: “Overlapping confidence intervals between the handheld devices indicated no significant difference of task completion of Step 4 between specific devices (Table 3).” - this is not shown in Table 3, please add it to Table 3, or to supplementary material. It’s also possible for a difference to be statistically significant when the overlap is small – please see this paper: https://www.cmaj.ca/content/cmaj/166/1/65.full.pdf

Please test the difference between the groups to see if it is significantly different from zero rather than just compare the confidence intervals of each group.

8. Efficiency: please include these results as a table in the paper. Also important to note that some children may not be measurable because they are very sick (have low perfusion) – studies have shown an association between inability to get a reading and mortality outcomes (e.g. your Ref 11, and Hooli et al 2020 AJTMH: https://www.ajtmh.org/view/journals/tpmd/102/3/article-p676.xml?tab_body=pdf ). Do you have an idea of how many children this might have been the case for? Also were the same children tested with all five devices? If not, then the mix of children may have been behind differences between devices. Table S1 shows prevalence of hypoxaemia is high (22%). This suggests quite a sick population (though could be due to altitude in some settings as you mention in your discussion)

6. PLOS authors have the option to publish the peer review history of their article (what does this mean?). If published, this will include your full peer review and any attached files.

**Do you want your identity to be public for this peer review?** For information about this choice, including consent withdrawal, please see our Privacy Policy.

Reviewer #1: **Yes: **Arjun Chandna

Reviewer #2: **Yes: **Dr Solomon Hailemariam Tesfaye

Reviewer #3: No

Reviewer #4: **Yes: **Tim Colbourn

---

## [Decision Letter · Decision Letter 1]

28 Apr 2023

PGPH-D-22-02104R1

Usability of pulse oximeters used by community health and primary care workers as screening tools for severe illness in children under five in low resource settings

A cross-sectional study in Cambodia, Ethiopia, South Sudan, and Uganda

Dear Dr. Pfurtscheller,

Thank you for submitting your manuscript to PLOS Global Public Health. After careful consideration, we feel that it has merit but does not fully meet PLOS Global Public Health’s publication criteria as it currently stands. Therefore, we invite you to submit a revised version of the manuscript that addresses the points raised during the review process.

We look forward to receiving your revised manuscript.

Kind regards,

Sakib Burza, MBChB, MRCGP, MSc, PhD

Academic Editor

Journal Requirements:

Additional Editor Comments (if provided):

Many thanks for this much improved manuscript, and apologies for the delay in getting the feedback to you. There are a few minor issues that have been raised, once these have been addressed we will be happy to accept this.

Reviewers' comments:

Reviewer's Responses to Questions

**Comments to the Author**

1. If the authors have adequately addressed your comments raised in a previous round of review and you feel that this manuscript is now acceptable for publication, you may indicate that here to bypass the “Comments to the Author” section, enter your conflict of interest statement in the “Confidential to Editor” section, and submit your "Accept" recommendation.

Reviewer #1: (No Response)

Reviewer #2: All comments have been addressed

Reviewer #4: (No Response)

2. Does this manuscript meet PLOS Global Public Health’s publication criteria? Is the manuscript technically sound, and do the data support the conclusions? The manuscript must describe methodologically and ethically rigorous research with conclusions that are appropriately drawn based on the data presented.

Reviewer #1: Yes

Reviewer #2: Yes

Reviewer #4: Yes

3. Has the statistical analysis been performed appropriately and rigorously?

Reviewer #1: No

Reviewer #2: Yes

Reviewer #4: Yes

4. Have the authors made all data underlying the findings in their manuscript fully available (please refer to the Data Availability Statement at the start of the manuscript PDF file)?

Reviewer #1: Yes

Reviewer #2: Yes

Reviewer #4: Yes

5. Is the manuscript presented in an intelligible fashion and written in standard English?

Reviewer #1: Yes

Reviewer #2: Yes

Reviewer #4: Yes

6. Review Comments to the Author

Reviewer #1: I thank the authors for their detailed and considered replies to my comments and those of the other reviewers. The manuscript is much improved. Whilst I am still very supportive that this manuscript should be published, I do still have comments that I would like to see addressed first. Some of these comments do require additional analyses, which I recognize may be frustrating for the authors, but I believe these are important in order for the findings to be interpreted correctly by the readers (particularly the handling of missing data and exploration of any interplay between probe selection and age category in driving the effectiveness findings). My comments are listed in the order I re-read the manuscript. Line numbers refer to the tracked changes version of the manuscript.

1. I could not see the STROBE checklist – perhaps this was in another part of the submission or omitted by mistake?

2. I am in agreement with Reviewer 3 that the overall tone of the manuscript should perhaps be moderated with regards the positivity of the findings, in particular with regards the usability of inaccurate (and not recommended) finger-tip devices. In Line 83, please clarify that the satisfactory performance of the pulse oximeters relates to handheld devices and that finger-tip device performance was poor in the cited study (reference 15), which concluded that they should not be used in children under the age of five. Perhaps it would also be worth justifying why the authors think it is helpful to report usability of inaccurate and non-recommended devices? I can see (and agree) that probably this is because the data have been collected and results should be presented (partly to avoid repetition by other groups) but it might be helpful to explain that their assessment of usability of the finger-tip devices does not imply a change in the position of the authors and they (still) do not recommended the use of finger-tip devices in children under the age of five.

3. Related to the above, Lines 110-111 suggest that the performance of the finger-tip devices was deemed satisfactory, whereas I think actually only the performance of the handheld devices was deemed satisfactory (reference 15). Perhaps the authors are referring to the fact that all five devices had passed laboratory testing? In line with the above point, this should be clarified, to avoid confusion amongst readers that the authors are now recommending the finger-tip devices on the basis of satisfactory performance?

4. I note the comments from Reviewer 4 with regards missingness along with my earlier comments. I understand that multiple imputation may not have been appropriate due to missingness being concentrated in particular study locations but I am still not clear why listwise rather than pairwise deletion has been used? For participants/observations in which data on some (but not all) items in the checklist were available, why not include these data? The overall checklist completion rate could still be reported as it is currently. Is there a reason why data recorded for individual checklist items within incomplete checklists might be unreliable? If not, I cannot see a reason why pairwise deletion shouldn’t be used? I would at least want to see this approach as a sensitivity analysis to show it doesn’t change the conclusions, particularly as the authors spend much of the manuscript discussing and interpreting the overall findings of the study in the context of individual checklist item completion rates. Hence, it is very important to be sure that the estimates for individual checklist item completion rates are not biased by the way missing data have been handled.

5. I do not agree that it is not possible to explore whether the different completion rates in infants is driven by the difficulties in selecting the correct probe (Line 313) – if all (or a disproportionate amount) of incomplete checklists amongst infants is due to this item compared to older children (i.e. the association with age is confounded by device type [as probe selection only relates to the handheld devices]) that would support the author’s conclusions that a single-probe device would have a great advantage – and be reassuring (although admittedly perhaps not definitively so depending on power) that there is not some other factor at play with regards to why infants had lower completion rates. This is important because it lends credibility to the author’s overall recommendations that research and implementation of single-probe handheld devices should be prioritized. At the moment the tone of the discussion and analyses is that probe selection and age category are two separate issues: Line 273 – “also revealed”; Line 294 – “additionally”. However, I think the data are there to unpack this further to some degree.

6. Table 2 – missing frequency in the handheld devices row.

7. Line 309 – doesn’t make sense – do you mean “should be accelerated”?

8. Line 303 – as previously mentioned, I tend to agree with Reviewer 3 that generally there could be stronger messaging about the good usability findings of the finger tip devices being contextualized with their poor performance. The cited trial from your group actually recommends they are not used in children under five. Think this could be stated rather than simply saying the handheld devices performed better.

Reviewer #2: The authors addressed all the comments. I don't have additional comments.

Reviewer #4: Almost all of my comments have been addressed, however two issues still require further revision:

My response to your response to Reviewer 4 (me)

1. You wrote in your response: "The Satisfaction Score we built is a summated score that was subsequently standardized to a value between 0 and 100. Thus, if we would assume someone had rated all 19 Likert items with a score of 1 this would pertain to a Satisfaction Score of 20 ( (19*1)*100/(19*5) ). You can find this explanation in a shortened version in lines 185-190."

My new response: This means the score is actually between 20 and 100 as it’s not possible to score lower than 20 is it? Please correct methods line 187 to read “value between 20 and 100”. You could also add your explanation above to the paper to make it clear that the minimum score possible of all 1s would score 20 and the maximum possible score of all 5s would score 100.

2. You wrote in your response: "As you rightfully pointed out the missing data was concentrated in Uganda which also supports our conclusion that there were difficulties in the data collection process. The substantial amount of missing data limits the informative value of the results for Uganda since the sample is small."

My new response: Thanks – would be good to mention Uganda specifically on this point in the discussion – i.e. the sentence ending on line 347 could continue to say something like: “problems during the data collection process, especially in Uganda where most of the missing data was concentrated (Table S1)”

7. PLOS authors have the option to publish the peer review history of their article (what does this mean?). If published, this will include your full peer review and any attached files.

**Do you want your identity to be public for this peer review?** For information about this choice, including consent withdrawal, please see our Privacy Policy.

Reviewer #1: **Yes: **Arjun Chandna

Reviewer #2: **Yes: **Solomon H. Tesfaye

Reviewer #4: **Yes: **Tim Colbourn

---

## [Decision Letter · Decision Letter 2]

14 Jun 2023

Usability of pulse oximeters used by community health and primary care workers as screening tools for severe illness in children under five in low resource settings

A cross-sectional study in Cambodia, Ethiopia, South Sudan, and Uganda

PGPH-D-22-02104R2

Dear Dr.med. Pfurtscheller,

We are pleased to inform you that your manuscript 'Usability of pulse oximeters used by community health and primary care workers as screening tools for severe illness in children under five in low resource settings

A cross-sectional study in Cambodia, Ethiopia, South Sudan, and Uganda' has been provisionally accepted for publication in PLOS Global Public Health.

Best regards,

Sakib Burza, MBChB, MRCGP, MSc, PhD

Academic Editor

Please note the final comments from the reviewer:

Thank you for addressing the remaining comments so thoroughly and in particular for the additional inclusion of Table S3, which is very helpful.

I have no further substantive comments. I did notice that there may be a typo in the newly included Table S4, where the task completion rate for correct probe selection (Step 4) is given as 68.7% but the raw numbers appear to be 152/166 i.e. 91.6%. Given that the authors have indicated that these sensitivity analyses did not materially change the main findings I assume the typo is in the numerator and the completion rate of 68.7% is correct. If however the typo is in the percentage and the completion rate is actually 91.6% when using pairwise deletion then the authors may need to revise their statement in Lines 374/375 of the manuscript with regards the sensitivity analysis yielding similar results and not impacting the conclusions...

Congratulations to the authors on this really important piece of work.

Reviewer Comments (if any, and for reference):

Reviewer's Responses to Questions

**Comments to the Author**

1. If the authors have adequately addressed your comments raised in a previous round of review and you feel that this manuscript is now acceptable for publication, you may indicate that here to bypass the “Comments to the Author” section, enter your conflict of interest statement in the “Confidential to Editor” section, and submit your "Accept" recommendation.

Reviewer #1: (No Response)

2. Does this manuscript meet PLOS Global Public Health’s publication criteria? Is the manuscript technically sound, and do the data support the conclusions? The manuscript must describe methodologically and ethically rigorous research with conclusions that are appropriately drawn based on the data presented.

Reviewer #1: Yes

3. Has the statistical analysis been performed appropriately and rigorously?

Reviewer #1: Yes

4. Have the authors made all data underlying the findings in their manuscript fully available (please refer to the Data Availability Statement at the start of the manuscript PDF file)?

Reviewer #1: Yes

5. Is the manuscript presented in an intelligible fashion and written in standard English?

Reviewer #1: Yes

6. Review Comments to the Author

Reviewer #1: Thank you for addressing my remaining comments so thoroughly and in particular for the additional inclusion of Table S3, which is very helpful.

I have no further substantive comments. I did notice that there may be a typo in the newly included Table S4, where the task completion rate for correct probe selection (Step 4) is given as 68.7% but the raw numbers appear to be 152/166 i.e. 91.6%. Given that the authors have indicated that these sensitivity analyses did not materially change the main findings I assume the typo is in the numerator and the completion rate of 68.7% is correct. If however the typo is in the percentage and the completion rate is actually 91.6% when using pairwise deletion then the authors may need to revise their statement in Lines 374/375 of the manuscript with regards the sensitivity analysis yielding similar results and not impacting the conclusions...

Congratulations to the authors on this really important piece of work.

7. PLOS authors have the option to publish the peer review history of their article (what does this mean?). If published, this will include your full peer review and any attached files.

**Do you want your identity to be public for this peer review?** For information about this choice, including consent withdrawal, please see our Privacy Policy.

Reviewer #1: **Yes: **Arjun Chandna
